# Fluorination Treatment and Nano-Alumina Concentration on the Direct Current Breakdown Performance & Trap Levels of Epoxy/Alumina Nanocomposite for a Sustainable Power System

Muhammad Zeeshan Khan [1,2], Muhammad Shahzad Nazir [3,*], Muhammad Shoaib Bhutta [4,*] and Feipeng Wang [1]

1   School of Electrical Engineering, Chongqing University, Chongqing 400044, China; zeeshankhanee@cqu.edu.cn (M.Z.K.)
2   Department of Electrical Engineering, TUF, Faisalabad 38000, Pakistan
3   Faculty of Automation, Huaiyin Institute of Technology, Huai'an 223003, China
4   School of Automobile Engineering, Guilin University of Aerospace Technology, Guilin 541004, China
*   Correspondence: nazir@hyit.edu.cn (M.S.N.); shoaibbhutta@hotmail.com (M.S.B.)

**Abstract:** Epoxy resin is extensively used in gas insulated switches as a renewable energy coating due to its exceptional insulation, mechanical characteristics, and environmental friendliness. The higher resistivity of the epoxy resin causes numerous surface charges to accumulate on the surface of the epoxy resin as a result of carrier injection due to the high DC electric field, which may cause insulation failure of the power transmission system. In this study, various concentrations of epoxy resins blended with nano-alumina (nano-$Al_2O_3$) at 0 wt%, 1 wt%, 3 wt%, and 5 wt% were created. Afterwards, the epoxy resin and $Al_2O_3$ nanocomposites were fluorinated by utilizing a combination of $F_2$ and $N_2$ with a ratio of 20% $F_2$ at 0.05 MPa while maintaining the temperature at 40 °C. In order to improve dispersion, nano-$Al_2O_3$ was treated with a silane coupling agent called γ-aminopropyltriethoxysilane (KH550). Additionally, infrared spectroscopy based on the Fourier transform was used to investigate the structure of chemical bonds. Furthermore, the changes in the molecular chains were verified by the FTIR spectra. The DC breakdown strength of epoxy resin\$Al_2O_3$ nano-composites showed that breakdown strength significantly improved after gas-phase fluorination. Moreover, 1 wt% nano-$Al_2O_3$ showed a higher breakdown strength. The fluorinated layer had a charge-suppressing effect, reducing the charge injected into the polymer matrix of the epoxy-resin matrix and increasing its DC breakdown capability. Thermally stimulated current (TSC) measurements indicate that epoxy resin's trap energy and trap density are altered by nano-$Al_2O_3$ incorporation and fluorination treatment (gas-phase). It was also observed that introducing nano-$Al_2O_3$ at a lower concentration (e.g., 1 wt%) can hinder the growth of space charge in the polymer matrix of the epoxy resin, thus enhancing the deep traps' energy. Furthermore, a fluorination layer containing a strong polarization of C-F bonding would seize the charge injection from electrodes, thus decreasing the conductivity and suppressing the charge injection.

**Keywords:** gas-phase fluorination; nanocomposites; trap level; epoxy resin; breakdown

## 1. Introduction

More research has focused on epoxy resins than other insulating materials due to their exceptional electrical, mechanical, and thermal properties and their usage in producing renewable energy. Materials made of epoxy resin play a vital role for offshore wind farms due to their compatibility, low brittleness, light weight, and enhanced mechanical strength. Moreover, it has been an extensively used insulator matrix material in gas insulated switches [1]. In industry, nanoparticles are mostly incorporated into epoxy resins and other polymer matrices to improve the dielectric strength and decrease manufacturing costs [2]. Numerous studies have shown that an epoxy-resin matrix merged



with nanoparticles has excellent electrical properties, such as corona resistance, volume resistivity, and breakdown strength [3]. Regardless of these stimulating benefits however, nanoparticles exhibit higher surface energy and easily aggregate in a polymer matrix. They thus perform poorly electrically compared to tentative predictions. Therefore, the vital issue is to reduce the aggregation and ensure adequate nano-filler dispersion within the polymer matrix. M. Z. Khan et al. investigated the surface treatment of nano-$Al_2O_3$ particles incorporated in epoxy resin [4]. They observed that the modified nano-$Al_2O_3$ displayed excellent dispersion in the epoxy-resin matrix.

However, the insulator's intrinsic surface introduced by nano-$Al_2O_3$ particles may not be appropriate for some real-time industrial applications. For example, when an electric field is applied, the epoxy resin\$Al_2O_3$ nanocomposite body and surface are prone to charge accumulation because the space charge can be injected continuously for a long time. As a consequence, the DC breakdown strength of epoxy resins\nano-$Al_2O_3$ composites may be drastically lowered by exposure to an electric field of sufficient severity. Considering their excellent capacity for inhibiting charge buildup on a surface, there is a growing interest in surface modification techniques limiting flashover incidence in gas insulated switches.

The gas-phase fluorination of insulating materials has been commonly used in industry, as it is one of the variety of surface modification strategies that may be used to improve the polymer surface's electrical properties, achieving the desired results efficiently and economically [5]. The presence of C-F bonding on the surface layer of the polymer that is produced as a result of fluorination taking place in the gas phase offers the polymer-enhanced surface characteristics without having a detrimental effect on the bulk properties of the polymer [6–10]. Fukhra et al. [11] first synthesized elementary fluorine action by organic compound, which proved the feasibility of modifying organic compounds by fluorination. After the 1980s, fluorination technology became more important, and many countries worldwide started researching related fields. Khabashesku et al. [12] first studied the problems related to the fluorination of carbon nanotubes and significantly improved the dissolution of carbon nanotubes in organic solvents by fluorination technology. Anand [13] and Carstens [14] have researched polymer film materials' fluorinated treatment. These rapidly developing studies have revealed alterations in the chemical structure of the surface of the material before and after fluorination, which significantly promotes the development of gas-phase fluorination technology and widens its application. An investigation into the effect of fluorination treatment (gas-phase) on the flashover characteristics of an epoxy resin\$Al_2O_3$ nanocomposite was carried out by M. Z. Khan et al. [15] in air, vacuum, and $SF_6$ environments. They were able to demonstrate that the surface charges were neutralized following fluorination. Thus, the process of fluorination has been considered a practical approach to improving the insulating properties of epoxy resin-created dielectrics. Liu et al. [16] used fluorination to alter the surface layer of epoxy resin and they observed that 10 min fluorination can efficiently reduce the charge accumulation on the surface of epoxy resin. Mohamad et al. [17] studied the DC flashover behavior of fluorinated epoxy resins. It was observed that as the fluorinated epoxy-resin flashover increased, the surface charge accumulation was suppressed and the distortion of the electric field reduced.

In this investigation, nano-sized $Al_2O_3$ particles were treated with the silane coupling agent KH550 for the purpose of improving their ability to disperse throughout the epoxy-resin matrix. In order to hasten the anticipated surface degradation of the epoxy resin\$Al_2O_3$ nanocomposite, gas-phase fluorination was applied to the surface of the nanocomposites. In order to examine the chemical bonds present in the samples, Fourier transform infrared spectroscopy was used. In this investigation, we used an approach known as thermally stimulated current, or TSC, to investigate how gas-phase fluorination altered the charge trap characteristics of epoxy resin\$Al_2O_3$ nanocomposites. Following that, the samples were put through a DC breakdown strength test. Before and after 15 min, 30 min, and 60 min of fluorination, the DC breakdown capability of the epoxy resin\$Al_2O_3$ nanocomposite was determined using the parameters of the Weibull model. The findings indicated that gas-phase fluorination can significantly alter the chemical bonding and

surface structure of the nanocomposite. This is accomplished by increasing the degree to which $Al_2O_3$ nanoparticles are distributed uniformly throughout the epoxy resin. As a result, the DC breakdown performance is enhanced.

## 2. Experiment Details

### 2.1. Specimen Preparation

The diglycidyl ether of bisphenol (epoxy resin), curing agent methyl tetrahydro phthalic anhydride (MTHPA), and catalyst tri-Dimethylaminomethyl phenol (DMP-30) were combined in a proportion of 100:80:1 with mechanical agitation at 60 °C. After curing at 90 °C for 2 h, the temperature was raised to 110 °C for another 2 h to finish curing the epoxy resin infused with nano-$Al_2O_3$. When the curing process was complete, the specimens were brought down to 25 °C and then washed with distilled water and ethanol. After this, the specimens were dried in a vacuum at 40 °C to eliminate any remaining moisture. The molecular structures of DGEBA, MTPHA, DMP-30, and cured epoxy resin are shown in Figure 1.

**Figure 1.** Molecular structure of DGEBA, MTPHA, DMP-30, and cured epoxy resin.

### 2.2. Fluorination Treatment

Figure 2 shows the sketch for the gas-phase fluorination setup. The specific preparation process was as follows: firstly, the temperature in the tank was adjusted to 40 °C. Samples were cleaned with absolute ethanol, thoroughly dried, placed in a stainless steel tank, repeatedly filled with nitrogen, and discharged to remove oxygen and moisture in the tank. Then, a mixture of fluorine and nitrogen (20% $F_2$) was put into the tank, with a reaction pressure of 0.05 MPa [17]. After the reaction was completed, the internal gas was pumped out using an exhaust gas treatment system. Nano-$Al_2O_3$ samples were labeled as $F_x$ (0, 15, 30, or 60), representing fluorination time, and 0, 1, 3, and 5 wt% indicating the concentration of nano-$Al_2O_3$ in the epoxy-resin composites.

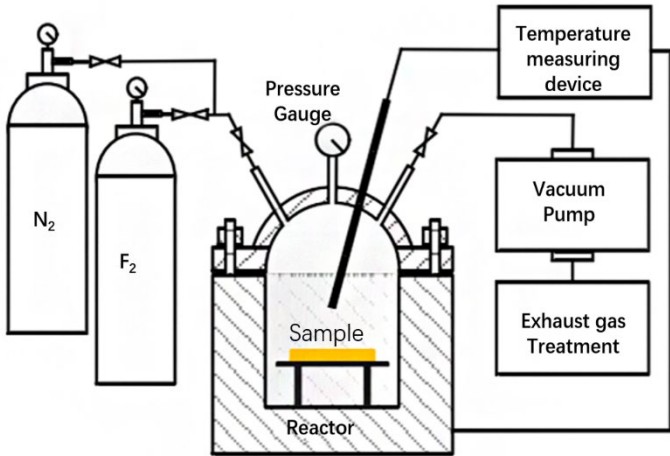

**Figure 2.** Sketch for experimental setup used for fluorination.

### 2.3. Experimental Techniques

The epoxy resin underwent chemical changes after being fluorinated. These changes were observed in the infrared and were verified using Fourier transform infrared spectroscopy (Alpha, Bruker). The thermal stimulated current of the specimens was measured by Novocontrol and data were recorded by its software. The measurement method of the depolarization current is shown in Figure 3. The measurement principle of the thermally stimulated current was as follows: the temperature ($T_1$) was made to rise to 90 °C; then an electric field of 333 V/mm was applied to polarize the sample, and the applied electric field was kept unchanged. The sample temperature was then reduced by 40 °C by $N_2$ gas to $T_2$, which reduced the kinetic energy charge.

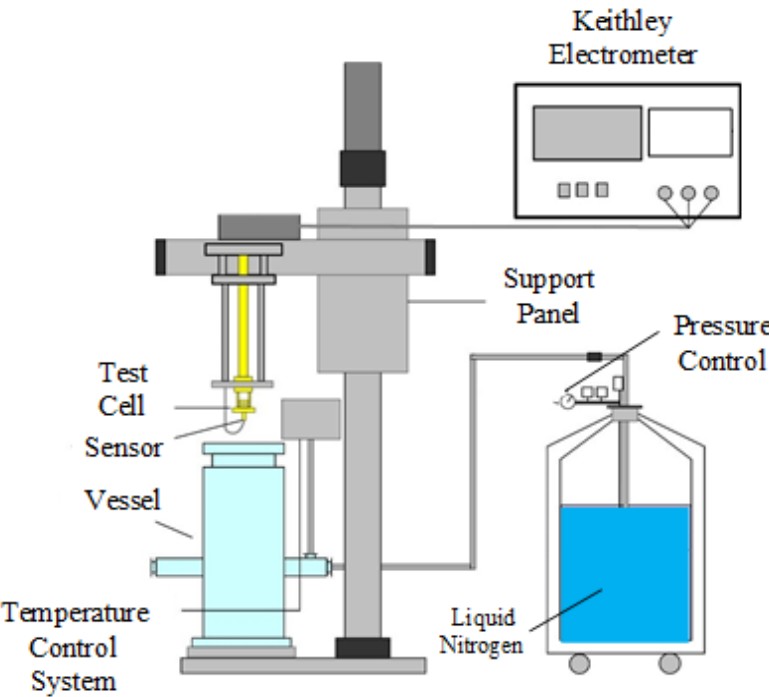

**Figure 3.** Sketch for experimental setup used for TSC measurement.

Moreover, the sample was always in the polarization state in this process. The external electric field was then removed to depolarize the sample, and the temperature was increased to 190 °C at the rate of 3 °C/min. As the temperature rose, the charge had enough kinetic energy to escape the trap and form the depolarization current, and the polarization

state gradually disappeared. An electrometer (Keithley 6517B) was used to measure the depolarization current.

The DC breakdown test was performed using a 25 mm diameter finger electrode system. The DC voltage increased at 0.5 kV/s until breakdown occured.

## 3. Results & Discussion

### 3.1. Fourier Transform Infrared Spectroscopy

The Fourier transform infrared spectra of the DGEBA, curing agent, accelerator, and cured epoxy resin are shown in Figure 4. It can be seen from the figure that absorption peaks at 1730 cm$^{-1}$ corresponding to C=O bond stretching vibration in the cured epoxy resin, and the absence of corresponding absorption peaks between acid anhydride (1777 and 1859 cm$^{-1}$) in the curing agent and epoxy group (915 cm$^{-1}$) in DGEBA indicates that the epoxy resin is cured more thoroughly [18].

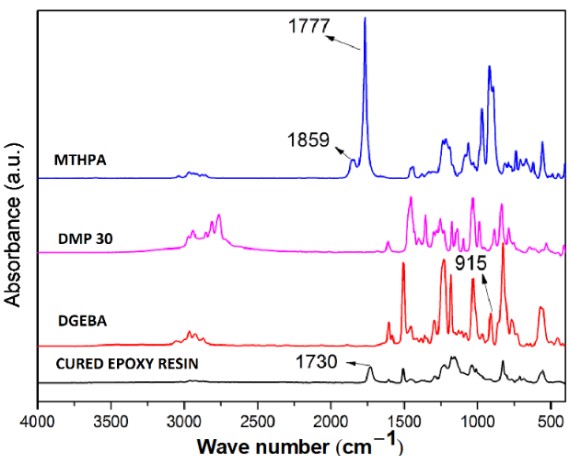

**Figure 4.** FTIR spectra of DGEBA, MTPHA, DMP-30, and cured epoxy resin.

The Fourier transform infrared spectra (FTIR) achieved for epoxy resin\Al$_2$O$_3$ nanocomposites before and after fluorination are depicted in Figure 5a. The fluorinated sample's concentration band at 2830–3000 cm$^{-1}$ disappears due to the hydrocarbon bond's stretching vibration in the aliphatic group. Furthermore, a significant drop in absorbance range was seen at 1730 cm$^{-1}$ on the fluorinated surface of the ester groups [18–20]. Due to the tensile vibration of the C=C bond in the aromatic ring 1-4-disubstituted benzene, the concentration peaks at 1508, 1042, and 827 cm$^{-1}$ were greatly attenuated in the fluorinated samples [19]. The robust concentration peaks related to CF, CF$_2$, and CF$_3$ groups were evident in extensive wave number ranges from 940 to 1300 cm$^{-1}$. The above observations confirm that the fluorination reaction changes the epoxy resin surface's chemical composition and bonding state by replacing hydrogen and incorporating a C=C bond reaction. Since the COF bond may define the degree of chemical bond breakage of epoxy resin during fluorination, a new absorption peak at 1810 cm$^{-1}$ (the hallmark of C-O-F groups) developed and exhibited a minor attenuation on the fluorinated epoxy resin/l$_2$O$_3$ nanocomposites. The effect of fluorination on the epoxy resin/Al$_2$O$_3$ nanocomposites at the absorption peak at 1810 cm$^{-1}$ is shown in Figure 5b. It can be observed from Figure 5b that with the extension of fluorination time, the intensity of the absorption peak corresponding to the COF bond increases obviously. In contrast, with the increase of the content of Al$_2$O$_3$ nanoparticles, the intensity of the absorption peak decreases slightly, indicating that long-time fluorination will aggravate the fracture of the molecular chain. At the same time, the nanocomposite can reduce the rupture of the molecular chain, but this effect will weaken with the increase of fluorination time.

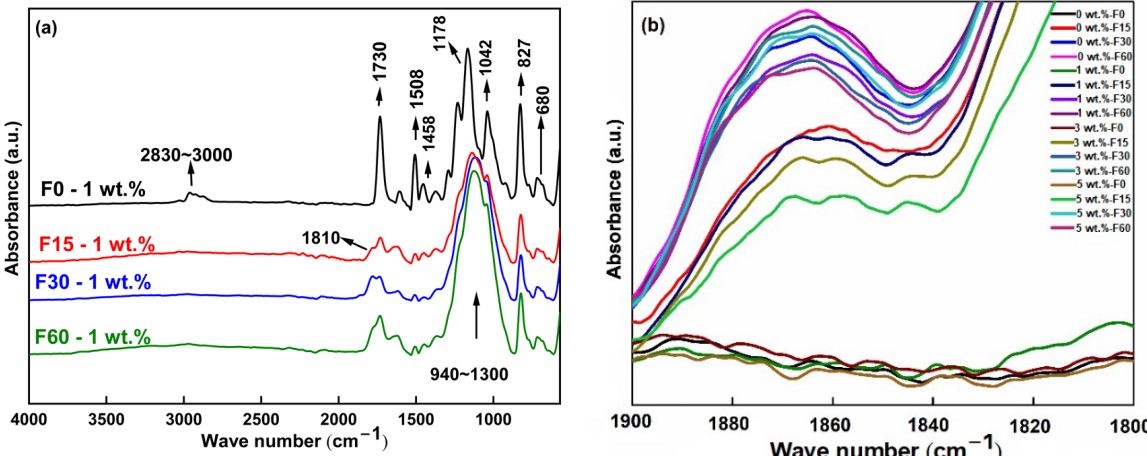

**Figure 5.** (**a**) FTIR spectra attained for 1 wt% $Al_2O_3$ epoxy resin composites using fluorination for 0 (F0), 15 (F15), 30 (F30), and 60 (F60) min; (**b**) Relationship between fluorination time/nanoparticle concentration and absorption peak intensity of COF bond in epoxy resin.

### 3.2. DC Breakdown Strength

To study the effect of nano-$Al_2O_3$ doping and gas-phase fluorination on the DC breakdown characteristic of epoxy resin/$Al_2O_3$ nanocomposites, a DC breakdown test was performed. The Weibull model of two parameters was performed to evaluate the breakdown strength of epoxy resin/$Al_2O_3$ nanocomposites before and after 15, 30, and 60 min of fluorination. The relationship between the DC breakdown strength at various nano-$Al_2O_3$ concentrations is shown in Figure 6 and Table 1. It can be observed from Figure 6 that the DC breakdown strength of the epoxy resin/$Al_2O_3$ nanocomposite enhances through nano-$Al_2O_3$ concentration.

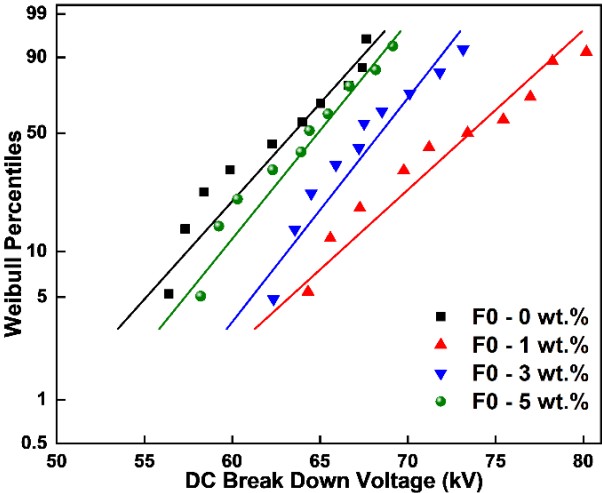

**Figure 6.** DC breakdown voltage of non-fluorinated epoxy resin\$Al_2O_3$ nanocomposites at various nano-$Al_2O_3$ concentrations.

**Table 1.** Weibull parameters of the DC breakdown strength of epoxy resin/$Al_2O_3$ nanocomposites.

| Samples | $\alpha$ (kV) | $\beta$ |
|---|---|---|
| F0—0 wt% | 60.55 | 43.01 |
| F0—1 wt% | 63.54 | 44.94 |
| F0—3 wt% | 66.62 | 60.76 |
| F0—5 wt% | 72.25 | 64.18 |

Additionally, the DC breakdown strength will be improved by the increase in trap energy caused by nano-$Al_2O_3$. Specifically, the trap features of the nano-dielectric are strongly influenced by the interaction zones [20]. At lower nano-$Al_2O_3$ concentrations (1 wt%), an independent interaction zone of $Al_2O_3$ nanoparticles creates deep traps and improves the breakdown capability of the nanocomposite. The overlap of the contact area increases the electron transport channel when the concentration of nano-$Al_2O_3$ rises up to 1 wt%. As the breakdown strength of the nanocomposites is reduced by the applied electric field, the charge carriers effectively conduct along these conduction routes. Wang et al. [21] investigated the dielectric properties of epoxy resin/$SiO_2$ nanocomposites after fluorination treatment. The breakdown capability of the epoxy samples was found to be significantly increased by fluorination treatment. On the other hand, the breakdown capability of nontreated epoxy-resin nanocomposites was lower than purified epoxy resin. However, the breakdown voltage of treated nanocomposites was improved at 3 wt%.

Figure 7 and Table 2 show the epoxy resin/$Al_2O_3$ nanocomposite breakdown strength after 15, 30, and 60 min of fluorination. The fluorinated nanocomposite's breakdown strength increased after fluorination for 15 min, and the epoxy resin\$Al_2O_3$ nanocomposites containing 1 wt% nano-$Al_2O_3$ still had the highest breakdown strength. Moreover, breakdown strength decreased after a long fluorination time, i.e., 30 and 60 min. This could be due to the following reasons. Gas-phase fluorination is an effective method of changing the surface properties without altering the epoxy-resin bulk properties [22,23]. The fluorinated layer has a charge-suppressing effect, which reduces the amount of charge injected into the polymer matrix of the epoxy resin and results in an increase in the breakdown capability. Moreover, long-term fluorination, e.g., from 30 to 60 min, will abolish the epoxy-resin matrix's symmetry, resulting in molecular chain fragments and decreased breakdown strength.

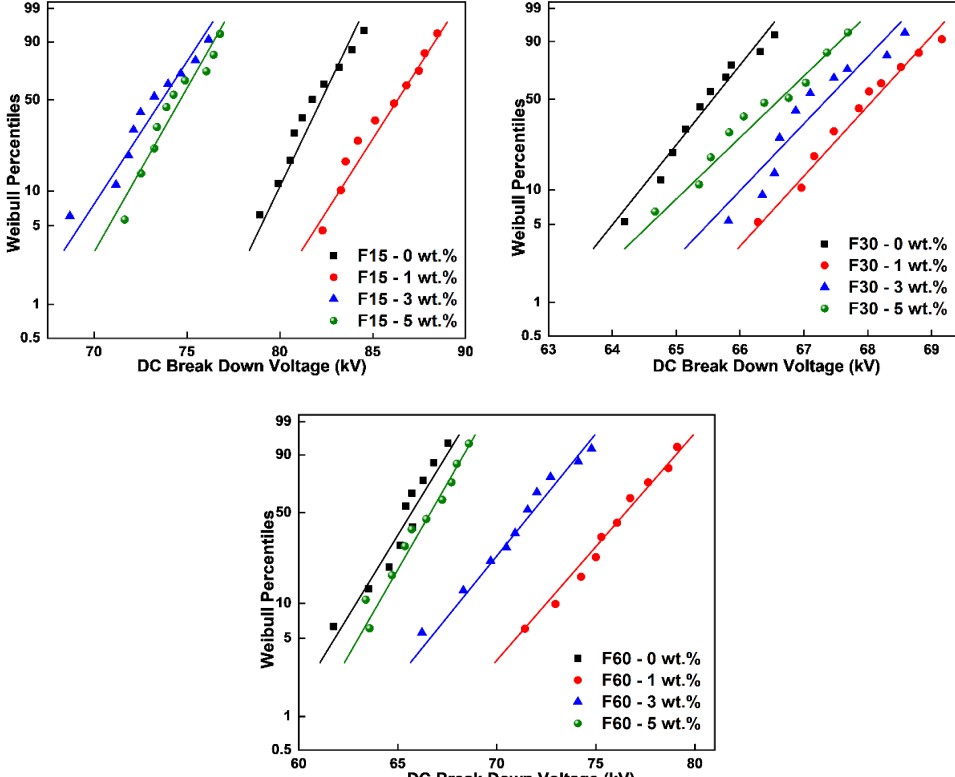

**Figure 7.** DC breakdown voltage of fluorinated epoxy resin/$Al_2O_3$ nanoparticles for 15 min (F15), 30 min (F30), as well as 60 min (F60) at different nano-$Al_2O_3$ concentrations.

**Table 2.** Weibull parameters for the DC breakdown strength of epoxy resin/$Al_2O_3$ nanocomposites before and after 15, 30, and 60 min of fluorination.

| Samples | Fluorination (F15) | | Fluorination (F30) | | Fluorination (F60) | |
|---|---|---|---|---|---|---|
| | $\alpha$ (kV) | $\beta$ | $\alpha$ (kV) | $\beta$ | $\alpha$ (kV) | $\beta$ |
| 0 wt% | 81.79 | 69.66 | 64.57 | 54.18 | 64.60 | 50.89 |
| 1 wt% | 84.78 | 67.24 | 65.46 | 56.97 | 65.91 | 53.07 |
| 3 wt% | 71.86 | 65.05 | 66.11 | 58.75 | 70.45 | 64.98 |
| 5 wt% | 73.66 | 64.70 | 67.96 | 59.19 | 74.64 | 65.94 |

*3.3. TSC Measurements*

Using the thermally stimulated current (TSC) technique, we studied how the charge-trapping capabilities of epoxy resin/$Al_2O_3$ nanocomposites are affected when nanoparticles and fluorination treatment are introduced. The MATLAB® application was used to compute TSC data. The equation below was used to extrapolate trap energy levels from current density charts [24,25].

$$j = ex^2/2d \int_{E_v}^{E_c} r_0(E)N(E)ve(-E/kT)e^{-1/B \int_{T_0}^{T} ve(-E/kT)dT} dE \qquad (1)$$

The order, density, temperature, Boltzmann constant, and escaping frequency of trapped electrons are all related to the sample thickness.

The equation below displays the dynamic status of the energy distribution as time passes. $A(E_m)$ is dependent on the anticipated analytical solution employed in present-day discharge theory [26].

$$r_0(E_m)N(E_m) = (2d/el^2)(J(T)/A(E_m)) \qquad (2)$$

Equations (1) and (2) were used to calculate trap energy levels from current density curves using TSC data.

Figure 8 presents data from the TSC that show that the trap density of epoxy resin\$Al_2O_3$ nanocomposites increases with nano-$Al_2O_3$ concentration. Moreover, epoxy resin incorporated with 1 wt% nano-$Al_2O_3$ has a higher trap energy. The zone of interaction that may be found within the polymer matrix of epoxy resin and nano-$Al_2O_3$ has been considered an essential factor that significantly influences an epoxy-resin nanocomposite's carrier mobility and electrical properties [26]. At a low nano-$Al_2O_3$ concentration, it is possible that the generated deep traps within the interface region will impede the mobility of the charge carriers, leading to a decrease in the trap density.

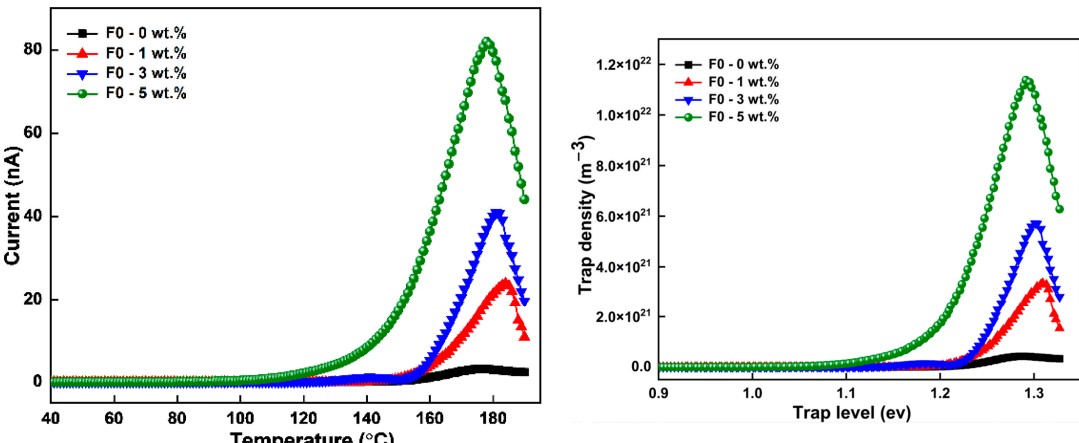

**Figure 8.** Trap energy level calculated from current density curves of non-fluorinated epoxy resin\$Al_2O_3$ nanocomposites at different nano-$Al_2O_3$ concentrations.

However, an increase of the nano-Al$_2$O$_3$ concentration may result in overlapping interaction regions. Conductive paths from overlapping interaction regions can decrease charge traps and increase trap density.

It can be seen from Figure 9 that trap density decreases with increasing time for the epoxy resin\Al$_2$O$_3$ nanocomposites fluorinated for 15, 30, and 60 min after washing. The following factors account for this. Fluorine possesses a high degree of electronegativity; because of this, the movement of surface charges that have accumulated on the epoxy resin\Al$_2$O$_3$ nanocomposites may be impeded. The surface conductance of the specimen is improved by fluorination, which creates shallow traps on its surface. Consequently, the depths of charge trap is significantly reduced for epoxy resin\Al$_2$O$_3$ nanocomposites fluorinated for 15 min.

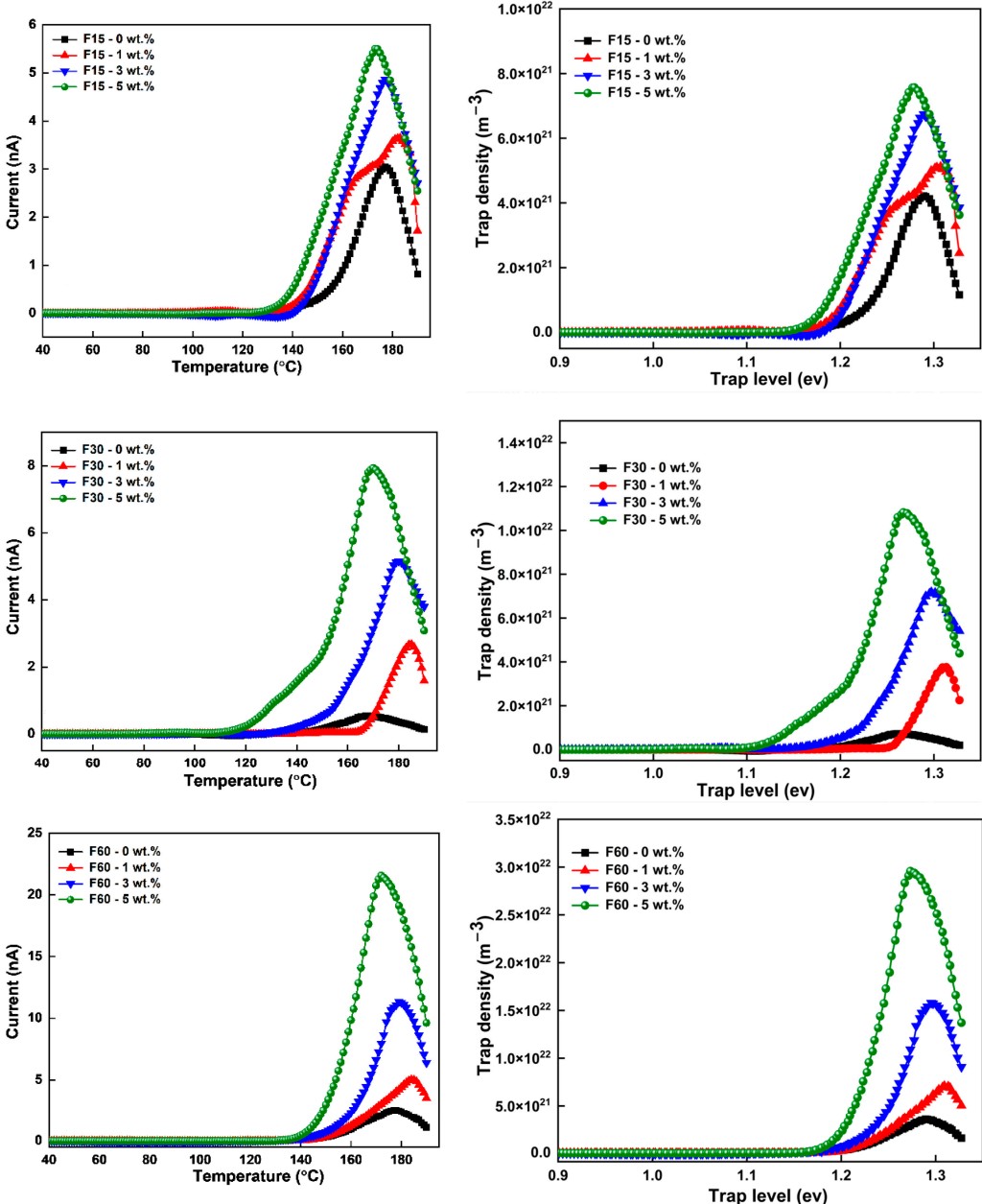

**Figure 9.** Trap energy level calculated from current density curves of epoxy resin\Al$_2$O$_3$ nanocomposites at different nano-Al$_2$O$_3$ concentrations fluorinated for 15 min (F15), 30 min (F30), and 60 min (F60).

FTIR analysis showed that the ester group at 1732 cm$^{-1}$ was reduced, and an increase in hydroxyl groups may also enhance shallow traps [27,28].

On the contrary, a long fluorination time, i.e., 30 and 60 min, leads to a slightly reduced trap energy level, which contributes to the destruction of the molecular structure and chain scissions. Table 3 shows the levels of trapped energy calculated from the current density curves of non-fluorinated and fluorinated epoxy resin\Al$_2$O$_3$ nanocomposites for 15 min (F15), 30 min (F30), and 60 min (F60) at different nano-Al$_2$O$_3$ concentrations. It can be observed that the trap energy level of the epoxy resin\Al$_2$O$_3$ nanocomposite initially increased at 1 wt% and later decreased at a higher concentration of nano-Al$_2$O$_3$. This comparison demonstrates that the energy level of both the shallow traps and the deep traps were located between 1.27 and 1.31 eV.

Additionally, it was noted that the trap energy level of 1 wt% nano-Al$_2$O$_3$ of non-fluorinated epoxy resin\Al$_2$O$_3$ nanocomposites was higher than the given fluorinated nanocomposites. Physical defects may be introduced on epoxy resin\Al$_2$O$_3$ as a result of physical alterations, particularly chain conformational changes during fluorination. Consequently, the overall depths of charge traps are gradually lowered as a result of fluorination.

After fluorination, C. Li et al. [29] analyzed the surface degradation of epoxy resin\alumina composites. It was observed that fluorination introduced shallow traps to the sample, and that the trap amount increased with increasing fluorination time.

**Table 3.** Trap energy levels estimated by current density curves of non-fluorinated and fluorinated epoxy resin\Al$_2$O$_3$ nanocomposites for 15 min (F15), 30 min (F30), and 60 min (F60) at different nano-Al$_2$O$_3$ concentrations.

| wt% | Time | 0 min (F0) | 15 min (F15) | 30 min (F30) | 60 min (F60) |
|---|---|---|---|---|---|
| 0 | | 1.27 | 1.28 | 1.26 | 1.28 |
| 1 | | 1.31 | 1.30 | 1.30 | 1.30 |
| 3 | | 1.30 | 1.28 | 1.29 | 1.29 |
| 5 | | 1.29 | 1.27 | 1.26 | 1.27 |

## 4. Discussion

A single scenario, represented in Figure 10, of generation and transport of space charge in epoxy resin/Al$_2$O$_3$ nanocomposites before and after fluorination, was designed to understand the epoxy's breakdown strength before and after fluorination. E is the electric field applied therein, and E$_{1'}$ and E$_{2'}$ are the negative electric potential closer to the electrodes. It can be seen that charge injection decreases after fluorination. This occurrence is due to the suppression effect of the fluorinated layer that results in reducing the magnitude of the space charge injection, thus reducing conductivity [30,31]. Moreover, incorporating nano-Al$_2$O$_3$ significantly influences the crystallization process and changes the microcrystalline morphology of epoxy, generating a large number of small uniform spherocrystals. The interface between the crystalline and amorphous regions also increased. Massive charges are fixed near the electrode interface, resulting in the accumulation of homo charges, thus expanding the interface's reverse electric field. Therefore, the electric interface field was significantly reduced, and the charges could not be quickly injected into the matrix, thus increasing the breakdown strength.

Furthermore, a large number of charges trap at these interfaces, which decreases the mobility of carriers. Therefore, very few charges could reach opposite electrodes and suppress space charge accumulation. Furthermore, neutralization and ionization phenomena co-exist, and low carrier mobility improves the neutralization process between negative and positive charges, thus weakening the impurity ionization [32]. As a result, hetero charges disappeared in epoxy resin\Al$_2$O$_3$ nanocomposites, thus enhancing the breakdown strength.

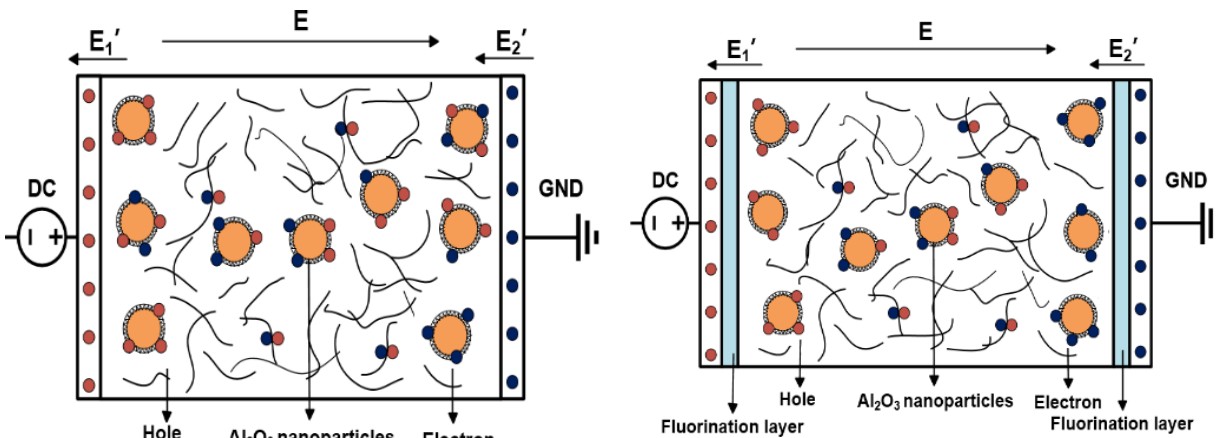

**Figure 10.** Generation and transport of space charge in epoxy resin\Al$_2$O$_3$ nanocomposites before and after fluorination.

## 5. Conclusions

To examine the impact of fluorination treatment on DC breakdown capability and trap levels in nanocomposites, in this study an epoxy resin containing nano-Al$_2$O$_3$ was gas-phase fluorinated. Based on the experimental results, the FTIR spectra showed that molecular chains were broken during the fluorination process. After being fluorinated in a gas phase, epoxy resin\nano-Al$_2$O$_3$ nanocomposites showed a dramatic increase in their DC breakdown strength. Furthermore, 1 wt% nano-Al$_2$O$_3$ showed a higher breakdown strength. It was concluded that the fluorinated layer has a charge-suppressing effect, reducing the charge injected into the epoxy-resin matrix and increasing the DC breakdown strength.

Thermally stimulated current (TSC) measurements indicated that epoxy resin's trap energy and trap density are altered by nano-Al$_2$O$_3$ incorporation and fluorination treatment (gas phase). Furthermore, it was noted that introducing the nano-Al$_2$O$_3$ at a lower concentration (e.g., 1 wt%) can hinder the accumulation of space charges in the polymer matrix of epoxy resin, thus enhancing deep traps' energy. Furthermore, a fluorinated layer together with strongly diverged C-F bonds increases apprehension of the charge inoculated from the electrodes, thus decreasing conductivity and suppressing the charge injection.

**Author Contributions:** Conceptualization, M.Z.K. and F.W.; methodology, M.S.B.; Matlab, M.S.N.; validation, M.Z.K., F.W. and M.S.B.; formal analysis, F.W.; investigation, M.S.B.; resources, M.S.N.; data curation, M.S.B.; writing—original draft preparation, M.Z.K.; writing—review and editing, M.S.B.; visualization, F.W.; supervision, M.Z.K.; project administration, F.W.; funding acquisition, M.S.N. All authors have read and agreed to the published version of the manuscript.

**Funding:** We are thankful to the Natural Science Foundation of China (NSFC) under Grant 51864014.

**Institutional Review Board Statement:** Not applicable.

**Informed Consent Statement:** Not applicable.

**Data Availability Statement:** The data of the current study can be obtained from the corresponding author.

**Conflicts of Interest:** The authors declare no conflict of interest.

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
