# Peer review of "Fluorination Treatment and Nano-Alumina Concentration on the Direct Current Breakdown Performance & Trap Levels of Epoxy/Alumina Nanocomposite for a Sustainable Power System"

_sustainability, doi:10.3390/su15075826_

Round 1
Reviewer 1 Report
Dear Authors, your study needs to minor revision. My comments was attached as a file.This manuscript written by Muhammad Zeeshan Khan et al. is a valuable research on. " Effects of Fluorination Treatment and Nano-Alumina Concentration on the DC Breakdown Performance and Trap Levels of Epoxy Resin/Alumina Nanocomposite ”. Even though the topic of manuscript is quite important, but the study is only acceptable after minör revision.
1. F2 and N2 combinations were used at a ratio of 1/4 in the abstract, but no reference was made for this ratio. Please explain in detail why this ratio was chosen.
2. 1% by weight Al2O3-nanocomposites showed 25% higher yields. Explain whether there is aggregation at 3% and 5%, and it would be more beneficial if rates such as 0% 0.5%-1%-1.5%-2% were preferred for higher fracture strength.
3. Chemical compounds need correction as sub-index, for example, Al2O3, N2....
4. There is no material section and at the same time the chemicals used in the experimental studies and their properties should be specified and explained.
5. Can you please formulate the time and temperature values when describing the experimental procedures, 2 hours shoul be 2 h,, 110 degree should be 110 0C….
6. at line 168, 827/cm ??? it should be cm-1, at the same time, it should be corrected as cm-1, that is, the upper index should be used for -1
7. Please provide reference for the comment made in the paragraph on lines 217-222
8. "Gas-phase fluorination is an effective method of changing the surface properties without altering the epoxy-resin bulk properties. Thefluorinated layer has a charge suppressing effect, which reduces the amount of charge injected into the polymer matrix of the epoxy-resin and results in an increase in the break down capability. Moreover, long-time fluorination, e.g., from 30 to 60 min, will abolish the epoxy-resin matrix's symmetry, resulting in molecular chain fragments and decreased breakdown strength."
9. It may be useful to include some literature on the electrical properties of nanocomposites in the introduction and results section.
Improvement of Synthesis and Dielectric Properties of Polyurethane/Mt-QASs+ (Novel Synthesis)
G Baysal, H Aydın, H HoÅŸgören, S Uzan, H Karaer, Journal of Polymers and the Environment 24 (2), 139-147
10. Add data table for Figure 4
11. there are hardly any references to comments made in the discussion section, please cite
12. The yields obtained in the result section should be added as numerical values.
13. some minor grammatical errors need to be corrected please check

Author Response
Type: Special Edition
Title: Effects of Fluorination Treatment and Nano-Alumina Concentration on the DC Breakdown Performance and Trap Levels of Epoxy Resin/Alumina Nanocomposite
Dear Editor,
Herewith we are resubmitting our revised manuscript for possible publication to Journal of Suistainability (MDPI). We are pleased that we have been given a chance at revising our manuscript. When performing the revision, we have done our best to address all the comments thoroughly and comprehensively. We have found them all very helpful for improving our manuscript. The manuscript has been carefully checked and revised according to the reviewers’ comments. Our replies and details regarding the revised manuscript are given in the continuation of this letter.
Yours sincerely,
Muhammad Zeeshan khan.
Detailed reply to Reviewers:
We thank the editor and reviewers for evaluating our manuscript. We very much appreciate the many insightful and helpful suggestions for improvement, and we have done our best to accommodate them all thoroughly. Our replies are as follows.
REVIEWER COMMENTS:
This manuscript written by Muhammad Zeeshan Khan et al. is a valuable research on. " Effects of Fluorination Treatment and Nano-Alumina Concentration on the DC Breakdown Performance and Trap Levels of Epoxy Resin/Alumina Nanocomposite ”. Even though the topic of manuscript is quite important, but the study is only acceptable after minör revision:
- F2 and N2 combinations were used at a ratio of 1/4 in the abstract, but no reference was made for this ratio. Please explain in detail why this ratio was chosen.
Author Reply:
Thanks for your’s valuable comments. Author has used 20% F2 by volume at 0.05 MPa for 15, 30, 45, and 60 min. Author has already performed experimentation on various ration of F2 i.e 10~50%. Gas-phase fluorination is beneficial to increase the surface conductivity of the epoxy resin, without compromising the bulk insulation properties. However, increasing ration will result in producing chemical defect that reduces the tensile strength and insulation properties of epoxy resin. At 20 % F2 by volume epoxy resin exhibit enhacned dielectric properties compare to pure epoxy resin. By Studying various review of literature mostly author used 20 % F2 by volume. Already incorporate refrence in manuscript.
- 1% by weight Al2O3-nanocomposites showed 25% higher yields. Explain whether there is aggregation at 3% and 5%, and it would be more beneficial if rates such as 0% 0.5%-1%-1.5%-2% were preferred for higher fracture strength.
Author Reply:
Thanks for your’s valuable comments. Nano-Al2O3 can produce many interaction zones in a epoxy resin at low filler concentration i.e 0.1 and 0.5%, the fraction of the extended loose polymer layers is high, which probably allows the existence of free ions and also their unhindered transport through the bulk of the material, causing a marginal increase in the electrical conductivity through the volume of the material. But as the nano-filler loading increases beyond 0.5%, the volume fraction of the extended loose polymer starts to decrease (increases the volume of immobile nanolayers). An increase in the fraction of immobile nanolayers in epoxy at slightly higher nano-filler loadings probably acts as ion traps which inhibit ion mobility resulting in the dc conductivity in the nanocomposite bulk to decrease and enhancing brealdown strength. For higher concentration i.e 3 wt.% and 5 wt.% more overlapping interaction zones were formed thereby gradually weakening the deep traps and the effect of the shallow traps gradually increased, and thus the decreased the volume resistivity and breakdown strength. Due to these reason author has choosen 1 wt.%, 3 wt.% and 5 wt.% by considering the various studies on epoxy nanocomposites at filler loadings of 1% and above [1]
[1] S. Singha and M. J. Thomas: Dielectric Properties of Epoxy Nanocomposites, IEEE Transactions on Dielectrics and Electrical Insulation, Vol. 15, No. 1;pp. 12-23, 2008
- Chemical compounds need correction as sub-index, for example, Al2O3, N2....
Author Reply:
Thanks for your’s valuable comments. Correction already been incorporated in manuscript.
- There is no material section and at the same time the chemicals used in the experimental studies and their properties should be specified and explained.
Author Reply:
Thanks for your’s valuable comments. Author has already incorporated the Molecular structure of DGEBA, MTPHA, DMP-30 and cured epoxy resin along with FTIR spectra of DGEBA, MTPHA, DMP-30 and cured epoxy resin to discuss chemical properties.
- Can you please formulate the time and temperature values when describing the experimental procedures, 2 hours shoul be 2 h,, 110 degree should be 110 0C….
Author Reply:
Thanks for your’s valuable comments. Correction already been incorporated in manuscript.
- at line 168, 827/cm ??? it should be cm-1, at the same time, it should be corrected as cm-1, that is, the upper index should be used for -1
Author Reply:
Thanks for your’s valuable comments. Correction already been incorporated in manuscript.
- Please provide reference for the comment made in the paragraph on lines 217-222
Author Reply:
Thanks for your’s valuable comments. Refrence has already been incorporated in paragraph lines 217-222
- "Gas-phase fluorination is an effective method of changing the surface properties without altering the epoxy-resin bulk properties. The fluorinated layer has a charge suppressing effect, which reduces the amount of charge injected into the polymer matrix of the epoxy-resin and results in an increase in the break down capability. Moreover, long-time fluorination, e.g., from 30 to 60 min, will abolish the epoxy-resin matrix's symmetry, resulting in molecular chain fragments and decreased breakdown strength." It may be useful to include some literature on the electrical properties of nanocomposites in the introduction and results section.
Improvement of Synthesis and Dielectric Properties of Polyurethane/Mt-QASs+ (Novel Synthesis)
G Baysal, H Aydın, H HoÅŸgören, S Uzan, H Karaer, Journal of Polymers and the Environment 24 (2), 139-147
Author Reply:
Thanks for your’s valuable comments. Refrence has already been incorporated in paragraph
- Add data table for Figure 4
Author Reply:
Thanks for your’s valuable comments. Table has already been added for figure 4
- there are hardly any references to comments made in the discussion section, please cite
Author Reply:
Thanks for your’s valuable comments. Refrence has already been incorporated in discussion section.
- The yields obtained in the result section should be added as numerical values.
Author Reply:
Thanks for your’s valuable comments. already been incorporated.
- some minor grammatical errors need to be corrected please check
Author Reply:
Thanks for your valuabe comments already revised and remove the grammatical mistakes.
---------------------------------------------------------------------------------------------------------------------

Reviewer 2 Report
This paper mainly studies the improvement of the electrical properties of epoxy resin / alumina nanocomposites, using fluorination treatment and nano-doping for surface modification and bulk modification. Please refer to the following recommendations :
1.In the introduction, there are a large number of literature statements on fluorination technology in various fields, but there are insufficient reports on the fluorination experiment of epoxy resin/alumina nanocomposites. At the same time, research reports on "nano modification" should be added, and a summary statement should be made.
2.In this study, epoxy resin / alumina nanocomposites with 0 %, 1 %, 3 % and 5 % concentrations were prepared. Will higher concentrations have better experimental results ?
3.In this paper, both surface modification and volume modification are used, will these two methods affect each other ? What is the basis for the fluorination time of 15,30 and 60 minutes ?
4.The infrared absorption peak mentioned in section 3.1 is shown in table 1, but table 1 is not found in the article.
5.Does the model established in Fig.10 have experimental or simulation data support ?
Author Response
Reviewer 2
This paper mainly studies the improvement of the electrical properties of epoxy resin / alumina nanocomposites, using fluorination treatment and nano-doping for surface modification and bulk modification. Please refer to the following recommendations:
- In the introduction, there are a large number of literature statements on fluorination technology in various fields, but there are insufficient reports on the fluorination experiment of epoxy resin/alumina nanocomposites. At the same time, research reports on "nano modification" should be added, and a summary statement should be made.
Author Reply:
Thanks for your valuable comments. Changes already been incorporated.
- In this study, epoxy resin / alumina nanocomposites with 0 %, 1 %, 3 % and 5 % concentrations were prepared. Will higher concentrations have better experimental results ?
Thanks for your valuable comments. Nano-Al2O3 can produce many interaction zones in a epoxy resin at low filler concentration i.e 0.1 and 0.5%, the fraction of the extended loose polymer layers is high, which probably allows the existence of free ions and also their unhindered transport through the bulk of the material, causing a marginal increase in the electrical conductivity through the volume of the material. But as the nano-filler loading increases beyond 0.5%, the volume fraction of the extended loose polymer starts to decrease (increases the volume of immobile nanolayers). An increase in the fraction of immobile nanolayers in epoxy at slightly higher nano-filler loadings probably acts as ion traps which inhibit ion mobility resulting in the dc conductivity in the nanocomposite bulk to decrease and enhancing break down strength. For higher concentration i.e 3 wt.% and 5 wt.% more overlapping interaction zones were formed thereby gradually weakening the deep traps and the effect of the shallow traps gradually increased, and thus the decreased the volume resistivity and breakdown strength. Due to these reason author has chosen 1 wt.%, 3 wt.% and 5 wt.% by considering the various studies on epoxy nanocomposites at filler loadings of 1% and above.
[1] S. Singha and M. J. Thomas: Dielectric Properties of Epoxy Nanocomposites, IEEE Transactions on Dielectrics and Electrical Insulation, Vol. 15, No. 1;pp. 12-23, 2008
- In this paper, both surface modification and volume modification are used, will these two methods affect each other ? What is the basis for the fluorination time of 15,30 and 60 minutes ?
Author Reply:
Thanks for your valuable comments. The biggest challenge of this research was to synthesize dielectric nanocomposites with various filler types. In order to make an organic polymer and inorganic filler compatible with each other, it is essential to create a chemical bonding between the constituents. To obtain a good dispersion of the nanoparticles, treatment with a coupling agent is a vital factor. Surface treated Al2O3, nanoparticles were incorporated into an epoxy resin to improve the particle dispersion. In addition Gas phase fluorination technology based on nano-composite was used to improve the surface conductivity to achieve high DC breakdown strength and improve dielectric properties.
The effect for the different fluorination time and surface roughness has been verified by Atomic Force Microscopy. It was observed that the surface roughness of the epoxy resin increases significantly with the prolongation of the fluorination time. Rough surfaces can greatly prevent the surface electron transfer resulting in high breakdown The effect of Fluorination on sample already been verified by author in his previous work by Atomic Force Microscopy as shown in fig. 1:
Fig. 1.1: Surface morphology of pure epoxy after fluorination for (a) 0 (F0), (b) 15 (F15), (c) 30 (F30), and 60 (F60) min.
It can be seen from the Figure 1.1, that the surface roughness of the epoxy resin increases significantly with the prolongation of the fluorination time. Rough surfaces can greatly prevent the surface electron transfer resulting in high breakdown voltage. Moreover, fluorination is a severe exothermic reaction. A large number of chemical bond replacement and fracture during the reaction process will significantly increase the surface roughness of epoxy resin, and the exothermic reaction will cause the stress difference between the surface areas and surface roughness’s of the sample rises significantly. However, Increasing fluorination time will result in chemical distortion thus reduce dielectric performance of epoxy resin.
Refrence:
[2] Muhammad Zeeshan khan, Aashir Waleed, Asim khan, Muhammad Arshad Shehzad Hassan, Zahir Javed Paracha and Umar Farooq. “Significantly Improved Surface Flashover Characteristics of Epoxy Resin/Al2O3 Nanocomposites in Air, Vacuum and SF6 by Gas-Phase Fluorination,” [J]. Electronic materials and science, vol. 49, pp. 3400-3408, 2020.
- The infrared absorption peak mentioned in section 3.1 is shown in table 1, but table 1 is not found in the article.
Author Reply:
Thanks for your valuable comments typo mistake already been corrected.
- Does the model established in Fig.10 have experimental or simulation data support ?
A single scenario, represented in Figure 10 of generation and transport of space charge in epoxy resin / Al2O3 nanocomposites before and after fluorination, was designed to understand the experimental data of epoxy's breakdown strength phenomenon before and after fluorination. However, we have also conducted simulation on comsol Multiphysics and it also support the mechanism of charge transport before and after fluorination. In this Manuscript Author has emphasis on experimental data.

Reviewer 3 Report
In this manuscript, authors have presented " Effects of Fluorination Treatment and Nano-Alumina Concen- 2 tration on the DC Breakdown Performance and Trap Levels of 3 Epoxy Resin/Alumina Nanocomposite". Thermally stimulated current measurements (TSC) indicate that 28 epoxy resin's trap energy and trap density are altered by nano- Al2O3 incorporation and fluorination 29 treatment (gas-phase).
Following corrections are required:
(a) " temperature is in- 136 creased to 190°C at the rate of 3°C / min". Is there any effect of heating rate? Briefly explain for choosing 3°C / min.
(b) A comparison table of previously reported work will be appreciated to present novelty of work.
(c) Discuss how the levels of trapped energy calculated from the Current Density Curves as shown in Table 4.
(d) Table 1 is missing in the manuscript.
Author Response
In this manuscript, authors have presented " Effects of Fluorination Treatment and Nano-Alumina Concentration on the DC Breakdown Performance and Trap Levels of 3 Epoxy Resin/Alumina Nanocomposite". Thermally stimulated current measurements (TSC) indicate that 28 epoxy resin's trap energy and trap density are altered by nano-Al2O3 incorporation and fluorination 29 treatment (gas-phase).
Following corrections are required:
- " Temperature is in- 136 increased to 190°C at the rate of 3°C / min". Is there any effect of heating rate? Briefly explain for choosing 3°C / min.
Thanks for your valuable comments. Heating will polarize the sample. The principle of thermal stimulation current is as follows: At first, when the temperature is T1, an electric field is applied to polarize the test sample and the applied electric field is kept unchanged. The sample temperature is then reduced to T2 quickly. In this process, the sample is always in the polarization state, and then the external electric field is removed and the sample is heated evenly. The sample is heated up because of the temperature rise. Moreover, 3°C / min is optimal rate for measuring the depolarization current, thus polarization state gradually disappears. The polarization process of the external electric field is the process of the electric field injected charge trapped by the trap in the sample. After reducing the temperature, the charge kinetic energy in the sample is reduced and it cannot be "de-trapped". As the temperature rises, the charge has enough kinetic energy to escape from the trap and form the depolarization current. By measuring the relationship between the depolarized current and the temperature of the sample, the curve of the sample temperature is the TSC characteristic curve, and the trap depth and the settlement density can be calculated by the relevant parameters in the curve.
- A comparison table of previously reported work will be appreciated to present novelty of work.
Thanks for your valuable comment. Author has already incorporated comparison of various filler concentration and Fluorination time in Table 1, 2 and 3. Moreover, Author has performed experiment more then 170 Samples to present a brief description related to optimal fluorination time and filler concentration to enhance the dielectric performance of epoxy resin.
- Discuss how the levels of trapped energy calculated from the Current Density Curves as shown in Table 4.
Thanks for your valuable comment. TSC tests of fluorinated and non-fluorinated epoxy /Al2O3 nanocomposite were carried out, and the parameters of trap energy levels from the TSC data are calculated using by a MATLAB® M file. The data collected from Alpha-A dielectric analyzer from NovoControl WinDeta Software. Then TSC data is incorporated in equations1 and 2 below to extrapolate trap energy levels from current density graph.
(01)
The order, density, temperature, Boltzmann constant, and escaping frequency of trapped electrons are all related to the sample thickness. Displays the dynamic status of the energy distribution as time passes. is dependent on the anticipated analytical solution employed according to the discharge theory [26].
(02)
4 Table 1 is missing in the manuscript.
Thanks for your valuable comments typo mistake already been corrected.

Round 2
Reviewer 2 Report
The reviewer believes that the paper can be accepted
Author Response
The reviewer believes that the paper can be accepted
Resposne:
Thanks for your appreciating comments.